# Cerebellar Metabolic Connectivity during Treadmill Walking before and after Unilateral Dopamine Depletion in Rats

**DOI:** 10.3390/ijms25168617

**Published:** 2024-08-07

**Authors:** Heike Endepols, Nadine Apetz, Lukas Vieth, Christoph Lesser, Léon Schulte-Holtey, Bernd Neumaier, Alexander Drzezga

**Affiliations:** 1Institute of Radiochemistry and Experimental Molecular Imaging, Faculty of Medicine and University Hospital Cologne, University of Cologne, 50937 Cologne, Germanylukas.vieth@uk-koeln.de (L.V.);; 2Nuclear Chemistry (INM-5), Institute of Neuroscience and Medicine, Forschungszentrum Jülich GmbH, Wilhelm-Johnen-Straße, 52428 Jülich, Germany; 3Department of Nuclear Medicine, Faculty of Medicine and University Hospital Cologne, University of Cologne, 50937 Cologne, Germany; alexander.drzezga@uk-koeln.de; 4German Center for Neurodegenerative Diseases (DZNE), 53127 Bonn, Germany; 5Molecular Organization of the Brain (INM-2), Institute of Neuroscience and Medicine, Forschungszentrum Jülich GmbH, Wilhelm-Johnen-Straße, 52428 Jülich, Germany

**Keywords:** behavioral PET, metabolic connectivity, cerebellum, locomotion, compensation, Parkinson’s disease

## Abstract

Compensatory changes in brain connectivity keep motor symptoms mild in prodromal Parkinson’s disease. Studying compensation in patients is hampered by the steady progression of the disease and a lack of individual baseline controls. Furthermore, combining fMRI with walking is intricate. We therefore used a seed-based metabolic connectivity analysis based on 2-deoxy-2-[^18^F]fluoro-D-glucose ([^18^F]FDG) uptake in a unilateral 6-OHDA rat model. At baseline and in the chronic phase 6–7 months after lesion, rats received an intraperitoneal injection of [^18^F]FDG and spent 50 min walking on a horizontal treadmill, followed by a brain PET-scan under anesthesia. High activity was found in the cerebellar anterior vermis in both conditions. At baseline, the anterior vermis showed hardly any stable connections to the rest of the brain. The (future) ipsilesional cerebellar hemisphere was not particularly active during walking but was extensively connected to many brain areas. After unilateral dopamine depletion, rats still walked normally without obvious impairments. The ipsilesional cerebellar hemisphere increased its activity, but narrowed its connections down to the vestibulocerebellum, probably aiding lateral stability. The anterior vermis established a network involving the motor cortex, hippocampus and thalamus. Adding those regions to the vermis network of (previously) automatic control of locomotion suggests that after unilateral dopamine depletion considerable conscious and cognitive effort has to be provided to achieve stable walking.

## 1. Introduction

It is well known that Parkinson’s disease has a long prodromal phase (lasting years or even decades) during which dopaminergic neurons gradually degenerate [1]. At the same time, numerous compensatory mechanisms develop which keep motor symptoms very light until compensation breaks down and the full symptomatology emerges [2,3,4]. Besides changes in the remaining dopaminergic system itself (such as increased dopamine metabolism, reduced dopamine transporter expression or alterations in dopamine receptors), changes in neuronal circuitry have been described [5]. There are numerous studies using resting state fMRI on Parkinsonian patients, but consistent functional connectivity changes have not been found [6]. There may be several reasons for this. First, different methods are used across research groups, and standard protocols have not been established [6,7]. Second, the disease is always progressing, and a stable situation where pathological and compensatory changes are balanced is never present. Third, there are individual differences between patients with respect to the regional severity and lateralization of dopamine depletion [8]. In addition, resting state functional connectivity may not be sufficient to describe the altered circuitry during the actual movement. Although it is not possible to let patients walk during an fMRI measurement, certain attempts have been made to overcome this drawback, such as gait simulation in virtual reality paradigms [9] or motor imagery [10]. Brain connectivity analyses based on EEG measurements during walking have been performed as well [11]. While all these efforts have produced invaluable results, two disadvantages of measuring human patients cannot be eliminated: the individual healthy baseline is hardly ever available and the distinct unilaterality of the disease is usually not found. This is the main strength of animal models, where the same animal can be measured before and after an experimentally induced lesion which can be located strictly unilaterally.

The unilateral (hemiparkinsonian) 6-hydroxydopamine (6-OHDA) rat model is one of the most common neurotoxin-based animal models for Parkinson’s disease [12,13]. A single injection of 6-OHDA into the lateral forebrain bundle of one hemisphere produces a well-defined dopamine depletion. The emerging phenotype is usually mild, because the unilateral nature allows for behavioral compensation from the healthy side [14]. This model is also suitable to investigate the compensation on the level of functional networks because, as a consequence of a single 6-OHDA injection, it reaches a stable phase where dopamine depletion does not progress any further. Due to these characteristics, we chose this unilateral model over a bilateral neurotoxin or genetic model, although the genetic models in particular can more accurately replicate Parkinson’s disease pathology [15,16]. The aim of our study was to compare walking-related functional networks in healthy rats (=baseline) with those in the stable chronic stage after unilateral 6-OHDA injection in the same animals.

We chose metabolic connectivity analysis using fluorodeoxyglucose ([^18^F]FDG)-PET, because this method can easily and repetitively be combined with locomotor activity. It works on the basis of metabolic trapping of [^18^F]FDG in tissues of high glucose consumption [17,18], allowing brain uptake and the accumulation of [^18^F]FDG in the conscious state during a behavioral task (in this case treadmill walking) followed by a PET scan under anesthesia. Although taking place after the behavioral task is finished, the PET scan captures the metabolic pattern established in the brain during the uptake period. Assuming that glucose consumption of functionally connected brain areas shows a mathematical correlation, the [^18^F]FDG-PET images can be used for seed-based connectivity analyses across a group of animals or patients. Metabolic connectivity analysis is an established method to study functional brain networks in humans [19,20] and animals [21,22,23]. It has already been used for examination in Parkinson’s disease patients [24,25,26,27,28] and in a 6-OHDA mouse model [29], but only in the resting state, which does not exploit the full potential of this method. In our study about walking-related networks, we were not a priori focused on a specific brain area, but rather took an exploratory approach where we assessed activation patterns during walking before and after unilateral dopamine depletion as a first step. Because the cerebellum was the area with the strongest increase of [^18^F]FDG uptake during walking compared to rest, we chose this brain region for the final metabolic connectivity analysis with two seed locations. As the cerebellum shows increased resting state-connectivity in Parkinson’s disease patients [30], our hypothesis was that the same would be found for walking-related connectivity in the rat model.

## 2. Results

### 2.1. [^18^F]FDOPA-PET

To assess the extent of 6-OHDA-induced dopamine depletion, we measured the accumulation of the radiolabeled dopamine precursor [^18^F]FDOPA semiquantitatively using the standardized uptake value ratio SUVR_Cer_ (i.e., intensity normalized to cerebellum, which is not affected in the 6-OHDA model [31,32]). The mean SUVR_Cer_ of the contralesional striatum was 1.34 ± 0.13, while the ipsilesional striatum had a significantly decreased mean SUVR_Cer_ of 1.05 ± 0.09 (*p* < 0.0001; Figure 1). Under the assumption that the contralesional striatum represents the healthy state (=100%) and that 1.00 is the cerebellum background value (=0%), [^18^F]FDOPA uptake decreased by 82 ± 20% in the ipsilesional striatum.

### 2.2. [^18^F]FDG-PET

We measured cerebral accumulation of the glucose tracer [^18^F]FDG semiquantitatively by expressing [^18^F]FDG uptake as SUVR_wb_, i.e., the whole brain mean value was set to 1. Because brain glucose consumption is mainly driven by synaptic activity [33], we will refer to [^18^F]FDG uptake as “activity” in the following paragraphs.

#### 2.2.1. Brain Activity during Treadmill Walking in Healthy Animals (=Baseline)

The most striking difference in brain activity between treadmill walking (ON) and treadmill switched off (OFF) was the high activity of the cerebellar vermis during locomotion, which was clearly visible in the average images (Figure 2A, marker 1). Voxel-wise statistical comparison revealed that activity significantly increased in the anterior part of the cerebellar vermis during treadmill ON. Other areas of activation were the ventral thalamus, hypothalamus, midbrain locomotor region, pons and inferior olive. The olivocerebellar tract was clearly discernable (for the full set of transverse section levels see Appendix A). In contrast, the cerebral cortex and striatum were more active during treadmill OFF (Figure 2A, markers 2 and 3). Statistical comparison demonstrated significant downregulation in parts of the striatum and almost all cortical areas except caudal parts of the primary motor cortex during walking. In addition, significantly higher activity during treadmill ON was found in the inferior colliculus (midbrain auditory nucleus; Figure 2A; IC) and the superior olive. This can be attributed to the sound of the moving treadmill and will not be further discussed.

#### 2.2.2. Changes in Brain Activity during Treadmill Walking after Unilateral 6-OHDA Dopaminergic Lesion

After unilateral dopamine depletion, almost all brain areas showed significant changes in brain activity compared to baseline, whereby decreases and increases were equally present. A few differences were already obvious by viewing the mean images: the activity in the ipsilesional striatum (=region with the highest dopamine depletion) was noticeably low (Figure 2B, marker 4). Furthermore, thalamic activation during walking was more pronounced bilaterally after the hemiparkinsonian lesion (Figure 2B, marker 5), which was confirmed by voxel-wise statistical comparison. Cerebellar activation increased as well, particularly in the posterior part of the vermis and the ipsilesional hemisphere (crus 2; Figure 2B, marker 6). For the full set of transverse section levels, see Appendix A.

The statistical comparison between baseline ON and 6-OHDA ON (Figure 2C) visualizes the significant walking-related activity changes caused by unilateral dopamine depletion: activity increased in the contralesional striatum and thalamus (particularly the medial thalamic nucleus), as well as in the cerebral cortex and cerebellar hemispheres (particularly ipsilesional) and the posterior cerebellar vermis. An extensive decrease in activity was found in hypothalamic, subthalamic and pontine regions. For the full set of transverse section levels, see Appendix A. The “difference of difference” is shown in Appendix A.

Separate paired t-tests are convenient for 3D datasets, because correction for multiple testing can be easily carried out. However, analysis of variance (ANOVA) would be a more appropriate statistical approach. For this reason, we extracted the SUVR_wb_-values of selected regions using volumes of interest (VOIs) and analyzed them with two-way repeated measures ANOVA followed by Sidak’s post hoc multiple comparisons test. The results are shown in Figure 3 and Table 1. The ANOVA results confirm the strong activation of the cerebellar anterior vermis during steady walking before and after 6-OHDA (Figure 3A). After 6-OHDA injection, the ipsilesional cerebellar hemisphere increased its activity during walking, but not when the treadmill was switched OFF (Figure 3B); the same was seen for the mediodorsal thalamus (Figure 3C). In contrast, the ipsilesional motor cortex M1 increased its activity during both regimes, ON and OFF (Figure 3D), while the activity of the contralesional hippocampus was slightly increased during treadmill OFF, but not during walking (Figure 3E). The contralesional striatum was equally active during treadmill OFF and ON after 6-OHDA injection, while a lower activity was observed during treadmill ON at baseline (Figure 3F). The activity of the ipsilesional striatum was decreased after 6-OHDA injection during both ON and OFF (Figure 3G), which reflects a direct effect of dopamine depletion.

#### 2.2.3. Changes in Brain Activity during Rest after Unilateral 6-OHDA Dopaminergic Lesion

The resting condition (treadmill OFF) is less well defined than the walking condition (ON) because the rats were free to roam as they wanted on the stationary belt. Nevertheless, the general effect of unilateral dopamine depletion was much clearer during OFF compared to ON. While in the ON condition, almost all brain areas displayed altered activity, while during treadmill OFF, the key regions became visible; a significant decrease in activity was found in the ipsilesional striatum, nucleus accumbens and olfactory tubercle, i.e., in the regions with initially the highest density of dopaminergic terminals (Appendix A). Decreased activity was also seen along the medial forebrain bundle (the fiber tract of nigrostriatal projections, where 6-OHDA injection had taken place) and in the ipsilesional substantia nigra region (where the cell bodies of dopaminergic neurons are located). This hypometabolism most likely reflects dopaminergic degeneration. Increased activity was found mainly in the contralesional hemisphere, in the primary and secondary motor cortex (middle layers), the mediodorsal and laterodorsal thalamus and the ventral hippocampus. A bilateral increase took place in the superior and inferior colliculus.

#### 2.2.4. Metabolic Connectivity of the Cerebellum during Treadmill Walking

In order to assess connectivity between brain areas during walking, seeds were placed in regions of interest, and correlation analyses with all other voxels of the brain were performed. The first seed was positioned in the anterior vermis of the cerebellum, the region of highest activation during treadmill ON (Figure 4A, marker 1). Correlation analysis revealed that the anterior vermis had very little connection with the rest of the brain, only showing a connection with the insula, the deep layers of the motor cortex and the ventrolateral periaqueductal gray (vlPAG). After the 6-OHDA lesion, the connectivity pattern of the anterior vermis (Figure 4B, marker 3) changed; it was now strongly connected to the middle and deep layers of the motor cortices (M1 and M2) as well as the dorsal hippocampus and the contralesional medial thalamus.

To investigate how the ipsilesional cerebellar hemisphere was connected during walking, a seed was placed at the spot of highest activation after the 6-OHDA lesion (in the region of the paramedian lobule and crus 2; Figure 4A, marker 4). For the comparison with the baseline condition, a seed was placed in the same location in the healthy animals (i.e., in the future ipsilesional cerebellar hemisphere; Figure 4A, marker 2). In healthy animals, the cerebellar hemispheres had approximately the same activity during both treadmill ON and OFF. Although not exceedingly active during walking, the (future) ipsilesional cerebellar hemisphere was nevertheless strongly connected to the rest of the brain. Positive correlations were found with all sensory and motor cortices (particularly ipsilateral), as well as with the contralateral medial thalamus and contralateral striatum. In addition, widespread negative correlations were found as well, in the ipsilateral nucleus accumbens and the contralateral insular, subthalamic, hypothalamic and pontine regions. After the 6-OHDA lesion, these connections were no longer present. The ipsilesional cerebellar hemisphere (Figure 4B, marker 4) was now connected to the vestibulocerebellum (nodulus and uvula) only. For the full set of transverse section levels, see Supplementary Appendix A.

## 3. Discussion

### 3.1. Brain Activity during Treadmill Walking in Healthy Animals

The most strongly activated brain area during treadmill walking in this work was the anterior vermis of the cerebellum. This brain region also stood out in previous rat treadmill studies using the regional cerebral blood flow (rCBF) markers [^14^C]iodoantipyrine and [^64^Cu]PTSM [35,36,37,38]. In human fMRI studies, the cerebellar vermis was found to be active during both real and imagined walking (for review see [39]). Other areas activated during walking in rats were the ventral thalamus, hypothalamus, pons and inferior olive. These regions are all well known to be involved in locomotion [40,41]. Surprisingly, most cortical areas were downregulated during walking, including all sensory cortices, and the frontal, cingulate and retrosplenial cortices. Only the caudal part of the primary motor cortex showed no significant difference between walking and rest. Because walking on a horizontal treadmill without any obstacles may engage mostly automatic processes (e.g., periodic limb movements, postural control), it is not surprising that the medial parts of the cerebellum (vermis) were most active. Together with the midbrain and subthalamic locomotor regions and the spinal locomotor network, the cerebellar vermis is involved in generating the locomotor rhythm [42]. In particular, the cerebellum is thought to predict the precise timing of stimuli related to movement, in the form of a forward controller [43,44,45]. Information used for timing is provided by the inferior olive, which is structurally connected to the cerebellum [46]. While the cerebellum ensures smooth and coordinated movement, the motor cortex and midbrain locomotor region are thought to initiate locomotion [41]. However, as shown in numerous studies with decerebrate cats (for review see [47]), the motor cortex does not seem to be essential, neither for the initial motor command nor for maintenance of locomotion on flat ground. It is rather involved in adjustments needed for skilled locomotion involving obstacles [48], where attention and voluntary control is required [49,50]. This function of the motor cortex may explain why, in contrast to our findings, a previous treadmill study with a similar setting [37] reported increased rCBF, particularly in the primary and secondary motor cortex and somatosensory cortex. While our rats were trained for three weeks, Wang et al. habituated their rats to the treadmill for four days only. Therefore, those rats were probably still in the learning phase, where treadmill walking was not yet fully automatic and still required considerable cortical control.

Another region which was downregulated during walking in our study was the striatum. As being the core part of the basal ganglia, the striatum is involved in motor decision-making and goal-directed movement, particularly in facilitating intentional movements and suppressing unwanted movement alternatives [51]. As treadmill walking is not goal-directed, and frequent decision-making is not required, it is conceivable that the basal ganglia were more active during treadmill OFF, where the rats were free to move on the stationary belt at their own choice (i.e., a mixture of walking, rearing, sniffing, grooming, etc.).

### 3.2. Brain Activity during Treadmill Walking after Unilateral 6-OHDA Lesion (Chronic Phase)

After the unilateral 6-OHDA lesion, we found significantly decreased activity in the ipsilesional striatum, the medial forebrain bundle (injection site) and the substantia nigra during treadmill OFF. This resembles our earlier study where we compared 6-OHDA-injected animals with a sham group [14] and this can be attributed to neuronal degeneration in the ipsilesional nigro-striatal pathway. Activity was increased bilaterally throughout the cortex (middle layers), retrosplenial cortex (deep layers) and colliculi (superior and inferior) and contralateral in the dorsomedial thalamus and ventral hippocampus. This pattern strongly intensified during treadmill ON. The clusters of decreased activity broadened and spread to the contralesional side, so that almost the whole ventral half of the brain was less active than before 6-OHDA lesion. Likewise, the number and size of the activated areas increased, and now also included the contralesional striatum, ipsilesional dorsal thalamus and bilateral cerebellum. We can therefore state that brain activation/deactivation caused by 6-OHDA became more symmetric during walking compared to rest. High bilateral cortical activation indicates that the cortical control of previously automatic locomotion increased not only for the lesioned side, but also for the healthy side. To that effect, the midbrain and subthalamic motor regions were less active in both hemispheres. Increased cortical and cerebellar activity has been described for Parkinsonian patients and interpreted as compensatory [52,53]. It has been suggested that cortico-cerebello-thalamo-cortical loops can compensate for the dysfunction of cortico-basal ganglia-thalamo-cortical loops [53], and our results support this hypothesis.

### 3.3. Metabolic Connectivity of the Cerebellar Vermis during Treadmill Walking

When a seed was placed in the most active area in the cerebellum during treadmill ON, the anterior vermis, hardly any functional connections with the rest of the brain were visible in healthy rats. It seemed that the high cerebellar activation was mostly intrinsic, and weakly correlated with only a few other brain areas (Figure 5A, yellow arrows). One of them was the insula, which is thought to be involved in sustained intentional movement with an emphasis on body ownership and self-agency [54]. The rat insula has been previously described to be functionally connected to the cerebellum [55], probably relayed via crossed connections to the vlPAG [56,57]. Although not traditionally part of the midbrain locomotor region [58], the vlPAG is nevertheless involved in the maintenance and coordination of locomotion, as well as in appetitive and defensive responses [41]. The synaptic connections of the vlPAG with the medial (fastigial) cerebellar nuclei of rodents have been described [59]. The metabolic connectivity between the cerebellum, vlPAG and insula can therefore be interpreted in terms of continuing locomotion. However, these correlations were weak, and we would expect a far more substantial network in the face of the extensive cerebellar activity during walking. Furthermore, it is highly unlikely that the cerebellum acts almost completely isolated from the rest of the brain. We therefore propose that the cerebellar vermis may switch functional networks regularly during walking. It has already been suggested that the same cerebellar structures may be involved in different connectivity patterns [60], and that the cerebellum is important for many different aspects of sensorimotor control, including cognitive functions [61]. If the cerebellar vermis is not committed to only one functional network, even in such a simple repetitive task as walking, significant correlations cannot be established on the basis of summed PET images. Using time-resolved fMRI data, a recent study has shown that the overall brain connectivity of humans was low during intervals of network switching [62]. Although in their study the cerebellum showed a low rate of network switching, this could be explained by the lack of (real or imagined) movement during the fMRI scan. Further studies are therefore needed to assess network switching of the cerebellar vermis during locomotion.

If we interpret the lack of substantial functional connections of the cerebellar anterior vermis during walking as flexible participation in numerous networks, the emerging connectivity pattern after the 6-OHDA lesion would indicate a weakening of cerebellar network switching. The bilateral connection of the anterior vermis with the dorsal hippocampus and primary motor cortex M1, as well as the connection with the contralesional medial thalamus (Figure 5B, yellow arrows), indicates a stable functional network highly suitable to compensate for unilateral dopamine depletion. As already described above, the motor cortex adds a voluntary component to a previously automatic task. In addition to the relay via the thalamus, direct connections between the motor cortex and cerebellar vermis have been described [63]. Furthermore, our results indicate that in the hemiparkinsonian rat model, the dorsal hippocampus participates in this network as well, in close association with the medial thalamic nucleus (Figure 5B, yellow arrows). The dorsal hippocampus is known for its “place cells” and spatial functions needed for sensorimotor integration [64]. Recently, hippocampal “time cells” have also been found, which code for the elapsed time of locomotion, while other neurons are influenced by the distance moved [65]. Hence, functional connectivity between the hippocampus and cerebellum has been described in tasks where spatial and temporal processing were important [66]. Furthermore, the hippocampus has been involved in motor compensation of caudate damage in Huntington’s disease [67]. The mediodorsal thalamic nucleus, a region assigned to cognitive activity [68,69], is active during spatial processing as well [70]. Although it is mostly known for its extensive connections to the frontal cortex [71], the mediodorsal thalamic nucleus is also a diencephalic hub involved in many functions by modulating the activity of numerous brain areas [72]. The hippocampus and mediodorsal thalamus are usually active during goal-directed behaviors, where they act in association with the prefrontal cortex [73]. Finding them in a walking-related network with the cerebellar vermis suggests that for the compensation of unilateral dopamine depletion, cognitive areas were recruited to support previously automatic behavior such as walking.

### 3.4. Metabolic Connectivity of the Cerebellar Hemisphere during Treadmill Walking

Our results revealed that after the 6-OHDA lesion, the ipsilesional cerebellar hemisphere showed increased activation during walking. Although in healthy animals activity of the future ipsilesional cerebellar hemisphere was the same during rest and walking, we placed a seed in the area of highest activation (paramedian/crus 2 region), seen later in the chronic phase after the 6-OHDA lesion. In contrast to the anterior vermis seed, the cerebellar hemisphere seed yielded an extensive network during walking in healthy animals (Figure 5A, light blue arrows). The cerebellar hemisphere was connected to the medial thalamus, striatum and numerous cortical areas, most likely relayed via the lateral and interposed cerebellar nuclei [74,75,76]. This is exactly the network described in a recent work, suggesting that the cerebellum participates in cognitive functions by modulating the coherence of neuronal oscillations [77]. Furthermore, the paramedian/crus 2 region can be activated via vibrissal stimulation [78] as well as fore- and hindlimb stimulation [79]. It has been suggested that this area is involved in perceptual prediction as well as in detecting sequential deviants [80]. It is therefore not surprising that it was equally active during both continuous walking and spontaneous movements during the resting state on the stationary belt.

In the chronic phase after unilateral 6-OHDA lesion, the cerebellar hemisphere (paramedian/crus 2 region) showed only one functional connection, which was to the vestibulocerebellum (nodulus and uvula). Again, it seems highly unlikely that a brain region increases its activity by more than 20% (with an effect size of 0.92 which is considered “large”) and at the same time disconnects from the rest of the brain besides one single region. Further work is needed to evaluate the possibility of network switching for the cerebellar hemisphere as well. With the available data, however, we can only discuss the connection to the vestibulocerebellum. The nodulus and uvula receive primary and secondary projections from the vestibular nerve and the vestibular nuclei [81]. They are involved in vestibulo-ocular reflexes and postural control [82,83], particularly in tilt discrimination [84]. It has been shown that stimulation in the nodulus/uvula evokes a response in the cerebellar paramedian lobule which is most likely mediated via parallel fibers [85]. Besides this direct connection, relay via the deep cerebellar nuclei would be possible as well [86] but unlikely since we did not see any functional connectivity of the hemispherical seed with the cerebellar nuclei. A functional connection of a region involved in perceptual prediction with an area crucial for postural control suggests a great effort to maintain proper balance during walking. We have already described in a previous work that hemiparkinsonian rats shift their weight to the ipsilesional hindpaw to unburden the affected contralesional forepaw during diagonal gait [14]. This may challenge lateral stability, which is thought to be the most critical aspect of postural control during locomotion [87]. The extensive connection of the cerebellar hemisphere with the vestibulocerebellum may therefore be interpreted as compensatory balance control.

### 3.5. Effects of Aging

Since the second phase of testing took place 6–7 months after 6-OHDA injection, some of the observed network changes may have been caused by aging. Most studies related to this topic have been carried out in humans. It has been demonstrated that in older adults hemispheric asymmetry decreases during cognitive [88] as well as motor tasks [89,90]. For a bimanual tracking task, reduced cerebellar connectivity (to both striatal and cortical areas) has been found in older individuals [91]. The same study reported reduced network flexibility associated with aging [91]. Movement initiation and execution networks undergo a decrease in connectivity between sensorimotor cortex (S1/M1) and striatum, while cerebellar connectivity is unchanged [92]. However, the cerebellum participates in motor networks during simple tasks (e.g., elbow flexion–extension) in old subjects, while young subjects perform the task without cerebellar involvement [93]. In rats, global structural and functional network features remain largely similar during aging [94]. However, connectivity decreases in sensory areas, while limbic connectivity increases [95]. During cognitive tasks, old rats engage larger networks compared to young rats [96]. Although locomotion networks have not yet been studied in rats, we can assume from these findings that age-related factors may have contributed to the increase in the vermis network in the chronic stage after 6-OHDA injection.

## 4. Materials and Methods

### 4.1. Experimental Design

We investigated how walking-related functional networks changed after a unilateral Parkinsonian lesion in the rat. We used a mild unilateral 6-OHDA model where gait changes are very subtle, indicating that the brain is able to almost fully compensate dopamine depletion [14]. As a first step, we assessed walking-related brain activation patterns in 17 healthy rats before 6-OHDA lesion (=baseline) using a horizontal treadmill. This was achieved by visualizing significant differences of [^18^F]FDG uptake during walking compared to rest. After 6-OHDA injection, we waited 4–6 months to measure unilateral dopamine depletion severity using 6-[^18^F]fluoro-L-3,4-dihydroxyphenylalanine ([^18^F]FDOPA)-PET to confirm that all animals had developed a substantial chronic hemiparkinsonian lesion. [^18^F]FDG-PET in relation to treadmill walking (same as baseline) was carried out 6–7 months after 6-OHDA injection in the same animals. We chose this long waiting period because we wanted the rats to be well advanced into the chronic state, where all possible inflammatory activity has been resolved and the remodeling of functional networks has been completed. Based on the activation patterns during walking, we selected two cerebellar areas (anterior vermis and cerebellar hemisphere) where we placed seeds for metabolic connectivity analysis. All procedures are described in detail below.

### 4.2. Animals

This study was carried out in accordance with the European Union directive 2010/63/EU for animal experiments and the German Animal Welfare Act (TierSchG, 2006) and was approved by the regional authorities, i.e., the State Agency for Nature, Environment and Consumer Protection (LANUV) of the State of North Rhine-Westphalia under the license number 84-02.04.2017.A310. Seventeen Long Evans rats (8 males + 9 females) participated in the experiments. At the beginning of the study, they were 4–5 months old and weighed 393 ± 36 g (males) and 264 ± 16 g (females). All animals were housed in groups of up to 4 animals in individually ventilated cages (NexGen Ecoflo, Rat1800; Allentown Inc., Allentown, NJ, USA) under controlled ambient conditions (22 ± 1 °C and 55 ± 5% relative humidity). Food and water for all animals were available ad libitum. The rats’ 12 h light/dark schedule was reversed (lights on from 9:00 p.m. to 9:00 a.m.) to allow treadmill walking and PET measurements to be in both the rats’ and the experimenters’ active phase (=dark phase for rats). Consequently, all behavioral experiments were carried out between 9:00 a.m. and 6:00 p.m. in the dark or under 660 nm red light.

### 4.3. Baseline Treadmill Training

The rats were trained for 3 weeks (3 sessions per week) on a rat treadmill (NG 47300-001, Ugo Basile, Gemonio, VA, Italy). Training lasted 35 min per session in the first week, 43 min in the second and 50 min in the third week. A training session started with 5 min on the stationary treadmill. Subsequently, the belt started moving with the slowest available speed of 3 m/min, which was maintained for 5 min. The speed was then gradually increased to 10 m/min over the next 15 min and maintained for the rest of the training session. The shock grid behind the moving belt was always switched off, because we did not want to induce fear. The rats were encouraged by gently pushing them back on the belt when they sat on the grid. After three weeks of training, all rats walked willingly for 45 min and needed no further encouragement to stay on the belt. With the final speed of 10 m/min, we aimed at the rat equivalent of human relaxed jogging, where the body moves effortlessly and almost automatically, while the conscious mind is only minimally engaged in the motor task. For rats, 10 m/min can be considered as a low speed (we therefore use the term “walking”). When using a running wheel voluntarily, laboratory rats move at approx. 25 m/min in short bouts of 2 min, with a total of 3 h per day [97]. They can reach a maximum speed of 36 m/min [98].

### 4.4. Surgery

After the baseline measurements were completed, the animals were anesthetized (initial dosage: 5% isoflurane in O_2_/N_2_O (3:7), reduced to 1.5–2.5% isoflurane for maintenance) and received 0.1 mL Carprofen (Rimadyl, Pfizer, Berlin, Germany) subcutaneously as pain relief. Each animal was fixed on a warming pad in a stereotactic system with a motorized stereotactic drill and injection robot (Robot Stereotaxic, Neurostar, Tübingen, Germany). For lesion generation, the skin was removed from a dorsal patch of the skull, and a small (approx. 1 mm diameter) hole was drilled through the bone. A Hamilton syringe was slowly inserted, and 21 µg 6-OHDA hydrobromide (stabilized with ascorbic acid; Sigma-Aldrich, Steinheim, Germany) in 3 µL NaCl were injected unilaterally into the medial forebrain bundle at the coordinates 4.4 mm posterior, 1.2 mm lateral and 7.9 mm ventral from Bregma. Injection side (left/right) was randomized. After injection, the syringe was left in place for 10 min to prevent spread of 6-OHDA through the injection canal. Subsequently, the syringe was slowly retracted, the trepanation was closed with bone wax (B. Braun, Melsungen, Germany), and the skin was sutured. Unilateral 6-OHDA injections are mild and do not affect the animals’ ability to take in water and food. Nevertheless, the animals were monitored daily, and all of them made a swift recovery without any behavioral abnormalities (e.g., rotation).

### 4.5. Treadmill Training after 6-OHDA Lesion

Retraining started 6 months after the 6-OHDA lesion. The body weights had increased to 514 ± 85 g (males) and 307 ± 26 g (females). The training resumed as described above with the difference in maintaining each training step for two weeks rather than one, leading to a total training time of 6 weeks. After the completion of training, all rats were able to walk for 45 min without any difficulties. Although subtle gait changes after this type of lesion are detectable using a gait analysis system [14], the rats did not suffer from obvious limping or any other motor abnormalities. For detailed gait analysis and correlation of gait parameters with dopamine depletion severity (measured with [^18^F]FDOPA-PET) and metabolic activity (measured with [^18^F]FDG-PET) in the same rat model, see [14].

### 4.6. PET Measurements

To confirm sufficient unilateral dopamine depletion, [^18^F]FDOPA-PET was conducted 4–6 months after 6-OHDA injection. As correlation between [^18^F]FDOPA uptake and dopaminergic cell death is well established [31,99,100,101], we did not perform histopathological examinations. For a typical example of tyrosine hydroxylase immunohistochemistry in this model, see [21]. Rats were briefly (for approx. 10 min) anesthetized with isoflurane in 30% O_2_ and 70% air (5% for anesthesia induction, 2% for maintenance) and a catheter (24 G Introcan Safety, B. Braun, Melsungen, Germany) was placed into the lateral tail vein. A total of 67.3 ± 5.1 MBq [^18^F]FDOPA (Forschungszentrum Jülich GmbH, Jülich, Germany) in 500 µL 0.9% NaCl was injected intravenously. The catheter was removed and the rats were transferred to their homecage for a 25 min awake uptake period. Subsequently, the rats were anesthetized again, placed on an animal holder (Minerve, Esternay, France) and fixed with a tooth bar in a respiratory mask. Their body temperatures were maintained at 37 °C using a feedback-controlled warming system. Their eyes were protected from drying out by the application of eye and nose ointment (Bepanthen, Bayer, Leverkusen, Germany). A PET scan in list mode was started 30 min after [^18^F]FDOPA injection and emission data were collected for 30 min, using a Focus 220 micro PET scanner (CTI-Siemens, Erlangen, Germany) with a resolution at the center of field of view of 1.4 mm. This was followed by a 10 min transmission scan using a ^57^Co point source for attenuation correction. The rats were returned to their home cage after the scan, where they woke up after a few minutes.

[^18^F]FDG-PET in combination with treadmill walking took place before (=baseline) and 6–7 months after 6-OHDA injection. On measurement days, the rats were briefly (for approx. 2 min) anesthetized, and 70.2 ± 2.9 MBq of [^18^F]FDG (Life Radiopharma, Bonn, Germany) in 500 µL 0.9% NaCl was injected intraperitoneally. The rats were immediately transferred to the treadmill, and the treadmill regime (“treadmill ON”) was started, which was identical to the last training session (i.e., 5 min rest, 5 min at 3 m/min, 15 min of gradual increase to 10 m/min, 25 min at 10 m/min). During the control regime, which was measured on another day, the treadmill remained switched off (“treadmill OFF”), and the rats spent 50 min on the stationary belt. After the [^18^F]FDG uptake period on the treadmill, the rats were anesthetized again and placed in the scanner. A PET scan in list mode was started 60 min after [^18^F]FDG injection and emission data were collected for 30 min, followed by a 10 min transmission scan. After the scans, rats were returned to their home cage.

### 4.7. Image Reconstruction and Postprocessing

After full 3D rebinning, the summed images were reconstructed using an iterative OSEM3D/MAP procedure, resulting in voxel sizes of 0.38 × 0.38 × 0.80 mm. For all further processing of the images including statistics, the software VINCI 5.21 for MacOS X (Max Planck Institute for Metabolism Research, Cologne, Germany) was used. [^18^F]FDOPA images were co-registered, whereby images with right-hemispheric lesions were flipped, so that the lesion was always on the left. They were intensity-normalized to the cerebellum because the cerebellum has very little dopaminergic innervation and therefore represents background [102]. For this, the images were divided by the mean value of the respective cerebellar VOI, resulting in the dimensionless “standardized uptake value ratio” SUVR_Cer_. The [^18^F]FDG images were co-registered (images with right-hemispheric lesions flipped) and intensity-normalized to the global mean, because the global brain energy budget varies only by 0.5 to 1.0% depending on stimuli from the environment [103]. To this end, each image was divided by the mean value of a VOI covering the whole brain, producing SUVR_wb_. For seed-based connectivity analysis, the [^18^F]FDG images were Gauss-filtered with 1 mm FWHM.

### 4.8. Image Statistics

To identify the brain areas involved in steady walking, the [^18^F]FDG images taken during “treadmill ON” were compared to the control regime “treadmill OFF” using a paired *t*-test. This comparison was carried out for both baseline the and after 6-OHDA lesion. To visualize changes induced by dopamine depletion, “treadmill ON” images taken after 6-OHDA injection were compared to baseline with a paired *t*-test. Resulting *p*-values were corrected for multiple testing using a threshold-free cluster enhancement (TFCE) procedure with subsequent permutation testing, thresholded at *p* < 0.05 [34]. TFCE and final thresholding were performed in VINCI using a python script. Permutation testing was carried out in RStudio 1.0 for MacOS X. In addition, the SUVR_wb_ values of selected brain regions were extracted through anatomically correct VOIs and analyzed with two-way repeated measures ANOVA using GraphPad Prism 8.4.3 for MacOS X. The two factors were “treadmill regime” (ON and OFF) and “lesion” (baseline and after 6-OHDA). ANOVA was followed by Sidak’s post hoc multiple comparisons test.

### 4.9. Metabolic Connectivity Analysis

Two seeds were placed in the cerebellum, one in the anterior vermis in the gray substance of lobule 2 and 3 (brain midline), the other in the ipsilesional cerebellar hemisphere in the region of the paramedian lobule and crus 2. Both seeds consisted of 4 voxels with a total volume of 0.46 mm^3^ per seed. The location of the anterior vermis seed was chosen because of the highest activation during walking in this brain area, and is the same as used in a previous study [38]. The hemispheric seed was placed in the area of highest hemispheric activation after 6-OHDA lesion. The mean seed values were extracted from every rat, and a Pearson correlation analysis of the seed with all other voxels of the brain was performed across all animals of one condition (“treadmill ON” baseline and after 6-OHDA, respectively, *n* = 17 each). The resulting correlation maps were corrected for multiple testing by TFCE as described above. This yielded four connectivity maps (2 seeds × 2 conditions) with brain voxels positively (displayed in red) or negatively (displayed in blue) correlated to the seed region. The positive correlations of brain voxels indicated that [^18^F]FDG uptake was similar to the seed; in animals where the seed uptake was high, the red-labeled voxels also had high uptake. Similar to the positive correlations in fMRI functional connectivity analysis, positive correlations in the metabolic connectivity analysis represent functional connections between brain areas [20,104]. Negative correlations indicated an [^18^F]FDG uptake inverse to the seed; in animals where the seed had high uptake, the uptake of blue-labeled voxels was low, and vice versa. Negative correlations are thought to reflect the regulatory interactions between brain regions, such as reciprocal modulations, suppression, inhibition and neurofeedback [105].

## 5. Conclusions

Taken together, our results revealed that walking-related cerebellar metabolic connectivity had changed fundamentally in the chronic phase after unilateral dopamine depletion. At baseline, the anterior vermis was strongly activated during walking, but showed hardly any stable connections. The cerebellar hemisphere, on the other hand, was not particularly active during walking, but was extensively connected to many brain areas including the motor and somatosensory cortex, the thalamus and the contralateral striatum. After unilateral dopamine depletion, the anterior vermis established a network involving the bilateral motor cortex, the hippocampus and the contralateral mediodorsal thalamus. The motor cortex and thalamus were taken over from the previous network of the cerebellar hemisphere, which strongly increased its activity during walking, but narrowed its connections down to the vestibulocerebellum after dopamine depletion. We have speculated that high neuronal activity combined with low connectivity may be an indicator of network switching, but this claim has to be substantiated through further research. The strong connection between the cerebellar hemisphere and the vestibulocerebellum most likely aids lateral stability during walking. It may allow the shifting of the main body weight away from the affected contralesional frontpaw to the ipsilesional hindpaw while maintaining an almost normal gait. Adding the motor cortex, hippocampus and mediodorsal thalamus to the vermis network of (previously) automatic control of locomotion suggests that after unilateral dopamine depletion, considerable conscious and cognitive effort has to be provided to achieve steady walking on a horizontal treadmill.

## Figures and Tables

**Figure 1 ijms-25-08617-f001:**
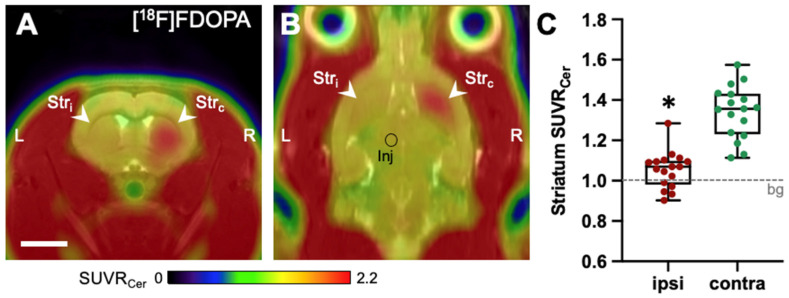
Unilateral dopamine depletion visualized using [^18^F]FDOPA-PET 4–6 months after 6-OHDA injection (*n* = 17). Images with lesion on the right (*n* = 8) were flipped to the left. (A + B): Average image, in transverse (**A**) and horizontal (**B**) orientation. (**C**): Box plots of mean tracer uptake in the ipsilesional and contralesional striatum (asterisk: paired *t*-test, *p* < 0.0001). Scale bar: 5 mm. Abbreviations: bg—background (=cerebellum), Inj—6-OHDA injection site (=medial forebrain bundle), L—left, R—right, Str_c_—contralesional striatum, Str_i_—ipsilesional striatum.

**Figure 2 ijms-25-08617-f002:**
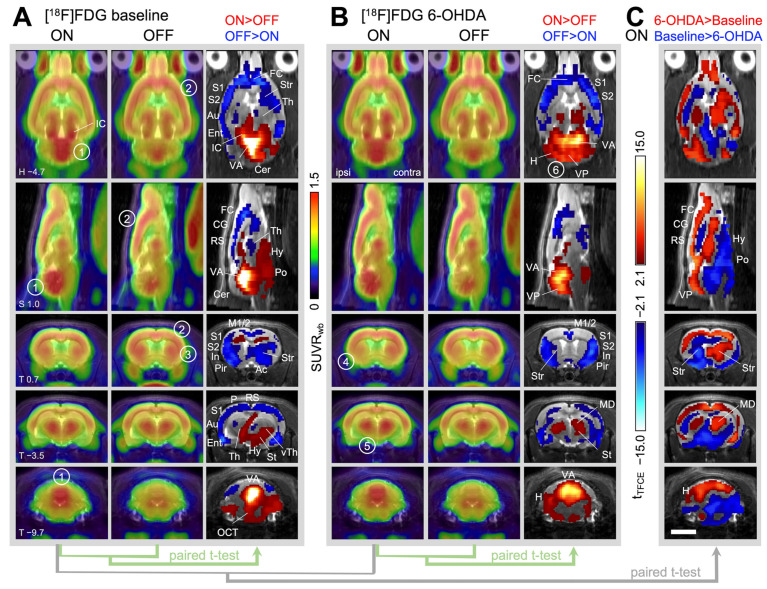
[^18^F]FDG uptake during walking compared to rest, before and after a hemiparkinsonian lesion (left side of images). Shown are horizontal (H; first row), sagittal (S; second row) and transverse sections (T; rows 3–5), where mean [^18^F]FDG images (*n* = 17) are projected onto an MRI template. Numbers in the lower left corner are coordinates in mm relative to Bregma. (**A**) Baseline, (**B**) 6–7 months after unilateral 6-OHDA injection. Lesion sides were randomized, right-sided lesions were flipped to the left. Columns “ON” comprise images from [^18^F]FDG during walking, while “OFF” shows [^18^F]FDG uptake when the treadmill was switched off. Both conditions were compared using a *t*-test, including correction for multiple testing (TFCE method followed by permutation testing [34]). Red voxels: higher activity during ON. Blue voxels: higher activity during OFF. (**C**) Statistical comparison (*t*-test) of [^18^F]FDG uptake during walking (treadmill ON) at baseline versus after 6-OHDA lesion. Red voxels: higher activity after 6-OHDA injection. Blue voxels: higher activity at baseline. Encircled numbers are markers for locating features described in the text. Scale bar: 5 mm. Abbreviations: Ac—Nucl. accumbens, Au—auditory cortex, Cer—cerebellum, CG—cingulate cortex, Ent—entorhinal cortex, FC—frontal cortex, H—cerebellar hemisphere, Hy—hypothalamus, IC—inferior colliculus, In—insular cortex, L—left (side of the lesion), M1—primary motor cortex, M2—secondary motor cortex, MD—mediodorsal thalamic nucleus, OCT—olivo-cerebellar tract, P—parietal cortex, Pir—piriform cortex, Po—pontine region, R—right, RS—retrosplenial cortex, S1—primary somatosensory cortex, S2—secondary somatosensory cortex, St—subthalamic region, Str—striatum, Th—thalamus, VA—cerebellar vermis, anterior lobe, VP—cerebellar vermis, posterior lobe, vTh—ventral thalamus.

**Figure 3 ijms-25-08617-f003:**
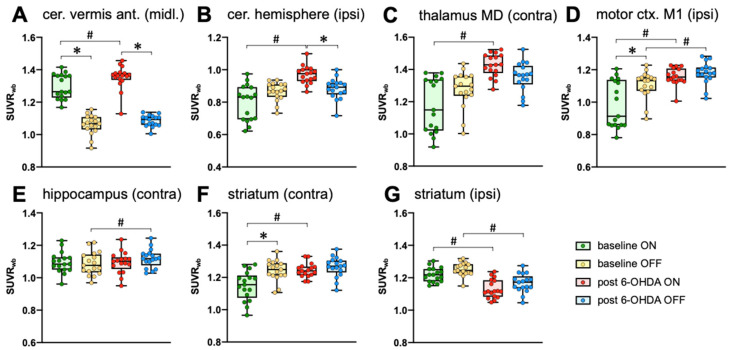
Two-way repeated measures ANOVA with [^18^F]FDG SUVR_wb_ values from selected VOIs: (**A**): cerebellar vermis, anterior lobe (midline), (**B**) ipsilesional cerebellar hemisphere (paramedian lobule and crus 2), (**C**) contralesional mediodorsal thalamus, (**D**) ipsilesional primary motor cortex, (**E**): contralesional dorsal hippocampus, (**F**) contralesional striatum, (**G**) ipsilesional striatum. * significant difference between treadmill ON and OFF. # significant difference between baseline and post 6-OHDA lesion. For full statistics, see Table 1.

**Figure 4 ijms-25-08617-f004:**
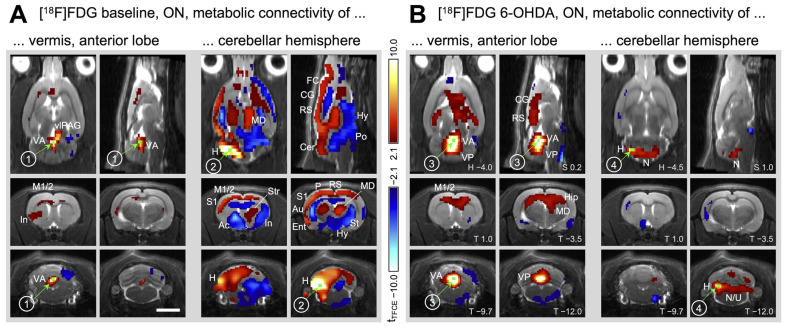
Metabolic connectivity of the cerebellum during walking before and after the hemiparkinsonian lesion (left side of images). Shown are horizontal (H), sagittal (S) and transverse (T) sections, where connectivity maps are projected onto an MRI template. Numbers in the lower right corner indicate coordinates in mm with respect to Bregma. Colored voxels display a significant correlation with the respective seed (green, labeled by an encircled number), corrected for multiple testing (TFCE method followed by permutation testing [34]). Red voxels: positive correlation with seed. Blue voxels: negative correlation with seed. (**A**) Baseline, (**B**) 6–7 months after 6-OHDA lesion. Scale bar: 5 mm. Abbreviations: Ac—Nucl. accumbens, Au—auditory cortex, Cer—cerebellum, CG—cingulate cortex, Ent—entorhinal cortex, FC—frontal cortex, H—cerebellar hemisphere, Hip—hippocampus, Hy—hypothalamus, In—insular cortex, L—left (side of the lesion), M1—primary motor cortex, M2—secondary motor cortex, MD—mediodorsal thalamic nucleus, MLR—midbrain locomotor region, N—nodulus, P—parietal cortex, Po—pontine region, R—right, RS—retrosplenial cortex, S1—primary somatosensory cortex, St—subthalamic region, Str—striatum, VA—cerebellar vermis, anterior lobe.

**Figure 5 ijms-25-08617-f005:**
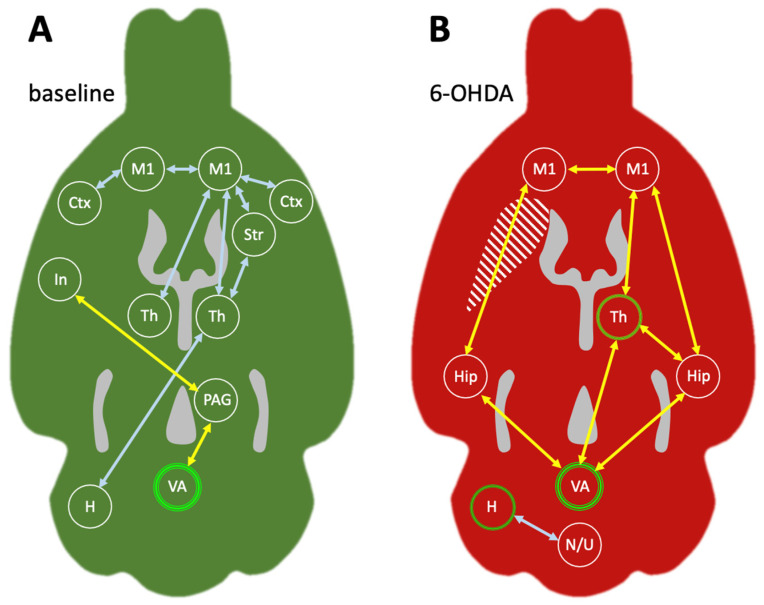
Metabolic connectivity of the cerebellar hemisphere (H; light blue arrows) and the cerebellar anterior vermis (VA; yellow arrows). This scheme is derived from the same animals (*n* = 17): (**A**) at baseline, i.e., in the healthy state; (**B**) in the chronic phase 6–7 months after unilateral 6-OHDA injection. Areas particularly active during walking are indicated by green circles. Areas with unchanged or decreased activation during walking are encircled in white. As a structural reference, the ventricles are included in gray. The dopamine-depleted striatum is indicated with a striped pattern. Abbreviations: H—cerebellar hemisphere (crus 2 and paramedian lobule), Hip—hippocampus, In—insula, M1—primary motor cortex, N—nodulus, PAG—periaqueductal gray, Str—striatum, Th—thalamus, U—uvula, VA—anterior vermis.

**Table 1 ijms-25-08617-t001:** Statistical results of two-way repeated measures ANOVA followed by Sidak’s post hoc test on regional [^18^F]FDG uptake, carried out separately for every VOI. Factors are “treadmill” (ON; OFF) and “lesion” (baseline; post 6-OHDA). Values are mean ± standard deviation (*n* = 17 per group). For group differences, effect size (Cohen’s d) and probability of type 1-error (*p*) are shown.

[^18^F]FDG SUVR_wb_	Cerebellum VA Midline	Cerebellum Hem. Ipsi	Motor Cortex M1 Ipsi	Hippocampus Contra	Striatum Ipsi	Striatum Contra	Thalamus MD Contra
Baseline ON	1.30 ± 0.08	0.80 ± 0.11	0.98 ± 0.14	1.09 ± 0.07	1.22 ± 0.04	1.15 ± 0.09	1.18 ± 0.17
Baseline OFF	1.06 ± 0.06	0.87 ± 0.06	1.11 ± 0.09	1.09 ± 0.07	1.25 ± 0.04	1.24 ± 0.06	1.27 ± 0.11
post 6-OHDA ON	1.35 ± 0.07	0.97 ± 0.06	1.16 ± 0.05	1.09 ± 0.07	1.13 ± 0.06	1.24 ± 0.04	1.43 ± 0.07
post 6-OHDA OFF	1.08 ± 0.03	0.88 ± 0.06	1.18 ± 0.07	1.12 ± 0.06	1.17 ± 0.06	1.26 ± 0.06	1.35 ± 0.09
main effect treadmill	**F(1,16) = 268.2** ***p* < 0.0001**	F(1,16) = 0.25 *p* = 0.6262	**F(1,16) = 17.9** ***p* = 0.0006**	F(1,16) = 1.1 *p* = 0.3017	**F(1,16) = 17.4** ***p* = 0.0007**	**F(1,16) = 23.0** ***p* = 0.0002**	F(1,16) = 0.4 *p* = 0.5226
main effect lesion	**F(1,16) = 6.0** ***p* = 0.0257**	**F(1,16) = 26.7** ***p* < 0.0001**	**F(1,16) = 31.1** ***p* < 0.0001**	F(1,16) = 2.3 *p* = 0.1478	**F(1,16) = 35.9** ***p* < 0.0001**	**F(1,16) = 22.3** ***p* = 0.0002**	**F(1,16) = 30.6** ***p* < 0.0001**
factor interaction	F(1,16) = 1.5 *p* = 0.2367	**F(1,16) = 15.4** ***p* = 0.0012**	**F(1,16) = 10.0** ***p* = 0.0061**	**F(1,16) = 5.2** ***p* = 0.0373**	F(1,16) = 0.005 *p* = 0.9464	**F(1,16) = 5.5** ***p* = 0.0327**	**F(1,16) = 9.8** ***p* = 0.0065**
Baseline ON vs. Baseline OFF	**d = 3.41** ***p* < 0.0001**	d = 0.78 *p* = 0.1071	**d = 1.04** ***p* = 0.0004**	d = 0.11 *p* = 0.8944	d = 0.81 *p* = 0.0511	**d = 1.21** ***p* < 0.0001**	d = 0.67 *p* = 0.0956
6-OHDA ON vs. 6-OHDA OFF	**d = 4.66** ***p* < 0.0001**	**d = 0.49** ***p* = 0.0320**	d = 0.33 *p* = 0.8171	d = 0.41 *p* = 0.0964	**d = 0.61** ***p* = 0.0425**	d = 0.37 *p* = 0.7284	d = 0.90 *p* = 0.2625
Baseline ON vs. 6-OHDA ON	**d = 0.74** ***p* = 0.0482**	**d = 0.92** ***p* < 0.0001**	**d = 1.60** ***p* < 0.0001**	d = 0.02 *p* = 0.9994	**d = 1.69** ***p* < 0.0001**	**d = 1.34** ***p* < 0.0001**	**d = 1.92** ***p* < 0.0001**
Baseline OFF vs. 6-OHDA OFF	d = 0.47 *p* = 0.7744	d = 0.28 *p* = 0.9245	**d = 0.92** ***p* = 0.0346**	**d = 0.49** ***p* = 0.0319**	**d = 1.72** ***p* < 0.0001**	d = 0.32 *p* = 0.7106	d = 0.76 *p* = 0.2097

Abbreviations: Contra—contralesional; Hem—hemisphere; ipsi—ipsilesional; MD—mediodorsal nucleus; VA—vermis anterior lobe. Bold indicates statistical significance (*p* < 0.05)

## Data Availability

The original data presented in the study are openly available in Jülich DATA at https://doi.org/10.26165/JUELICH-DATA/AD9X7K.

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
