# Peer review of "Cerebellar Metabolic Connectivity during Treadmill Walking before and after Unilateral Dopamine Depletion in Rats"

_ijms, 2024, doi:10.3390/ijms25168617_

Round 1

Reviewer 1 Report

Comments and Suggestions for Authors

The manuscript “Cerebellar metabolic connectivity during treadmill walking before and after unilateral dopamine depletion in rats” covers a scientifically attractive topic – metabolic connectivity and compensatory changes in the 6-OHDA-Parkinson's disease model. The manuscript is well organized, concise, presents the topic and results clearly, and I strongly believe that the manuscript will be of great interest to readers of the journal IJMS, after a few minor corrections and addressing of one major concern related to the age of the rats, which is discussed in detail below.

Abstract

The abstract is concisely written and clearly shows the idea and main findings of the study that explain the connectivity and compensatory changes that contribute to the maintenance of stable walking after a 6-OHDA lesion. I have no suggestions or corrections to this section.

Introduction

Like the abstract, the introduction was concisely written and included all relevant information, such as.: 1) the reasons for the still conflicting data on compensatory changes related to functional connectivity, 2) the rationale for using the animal model and its advantages over clinical studies, 3) the basic information about the 6-OHDA model that is useful to readers, who are not experts in Parkinson's models, 4) the basic information on [18F]FDG-PET and, most importantly, the highlighted gap in the literature (“it has already been used for examination in Parkinson's disease patients [22-26] and in a 6-OHDA mouse model [27], but only in the resting state, which does not exploit the full potential of this method”). I have no suggestions or corrections to this section.

Materials and Methods

Experimental design

The experimental design is well described and easy to understand.

Animals

The information in this subsection raises serious concerns when reading the study that need to be addressed.

1) Why are the rats 4-5 months old at the beginning of the study? Did the authors want to eliminate the difference between adult – at baseline measurement - and old rats at the end of the experiment?

2) The authors explained that they chose this long waiting period because they wanted the rats to be well advanced in the chronic state, where all possible inflammatory activity has subsided, and remodelling of functional networks is complete. However, I am pretty sure that neuroinflammation and network remodelling are completed earlier than 6 months after 6-OHDA-induced depletion of dopaminergic neurons. Please provide additional information on this topic.

3) How do the authors know that the observed changes in conectivity/activity are the result of only one factor (dopaminergic depletion) and are not influenced by a significant age difference? This aspect should be discussed in detail (further suggestions can be found below under discussion suggestions).

Results

The results are clearly presented and well described. I have no suggestions or corrections to this section.

Discussion

The authors provided a concise but interesting discussion of brain activity during treadmill running in healthy animals. They explained and discussed both activated regions such as the cerebellum, thalamus, etc., and even more interesting was the discussion of “down-regulated areas” such as sensory cortex, frontal, cingulate and retrosplenial cortex, and the striatum. However, the simplicity of the explanations and the main conclusions on brain activity during treadmill walking after unilateral 6-OHDA lesion and metabolic connectivity of the cerebellar hemisphere and vermis are highly commendable and this subsection shows that the authors really understand the results obtained and, more importantly, provides us, the readers, with a valuable “map” of activity before and after ipsilateral dopaminergic depletion.

 Despite all the praise for this work, however, the question of the age of the rats remains a major problem. The authors need to address the question of how they know that the observed changes in connectivity are the result of only one factor (dopaminergic depletion) and are not influenced by a significant age difference? This question should be addressed in a separate section such as “Limitations of the study” and include the data on old rats per se, the additional information on the need for a 6-month delay between 6-OHDA intoxication and the measurement of the before condition (literature data on neuroinflammation after 6-OHDA intoxication in MFB, i.e. the time profile of neuroinflammatory changes obtained from the available literature or the authors’ data, neural reorganization, etc.). Please provide these explanations and corrections to improve the relevance of the results obtained and the scientific transparency.

Author Response

Dear reviewer,

we are very grateful for your insights and valuable input to our manuscript. Thank you very much for your kind words - we greatly appreciated them. We have tried to address the issues raised, and we believe that in consequence the paper has been significantly improved.

Because all three points of criticism are related, we will address them together.

The age of the rats at the beginning of the study was not particularly important. We breed our Long Evans rats in-house from single pairs we buy once in a while from a certified breeder. So we usually use the rats we have in stock, and do not purchase them specifically for experiments. We only wanted the rats to be adult (not juvenile) and started the baseline experiments at a time convenient for us (i.e. no other experiments running, Master student available etc.). For the second phase after 6-OHDA lesion it was similar. We only wanted a long period of time between the lesion and the experiments (something between 4 and 8 months). We have done many studies about inflammation in the brain and found that inflammation still persisted after 6 months in a cardiac arrest model (Schroeder et al., 2021, DOI 10.1097/SHK.0000000000001546). In the 6-OHDA model we only looked during the first four weeks after 6-OHDA injection (unpublished results, but included in a dissertation [in German] here: https://kups.ub.uni-koeln.de/9481/1/Dissertation%20Stefanie%20Vus.pdf; Figs. 37–39). Some animals responded strongly with inflammation immediately after 6-OHDA injection, others didn't. The "responders" still showed greatly increased uptake of a neuroinflammation tracer (we used [18F]DAA1106 which binds to microglial TSPO) and in the corresponding immunohistochemistry a high number of Iba-1-positive microglia around the injection site after four weeks. Therefore we decided for the long waiting period.

But you are absolutely right in arguing that age might play a role in network changes, and that this issue must be discussed. We have added the following paragraph to the discussion:

"3.5. Effects of aging

Since the second phase of testing took place 6–7 months after 6-OHDA injection, some of the observed network changes may have been caused by aging. Most studies related to this topic have been done in humans. It has been demonstrated that in older adults hemispheric asymmetry decreases during cognitive [88] as well as motor tasks [89,90]. For a bimanual tracking task, reduced cerebellar connectivity (to both striatal and cortical areas) has been found in older individuals [91]. The same study reported reduced network flexibility associated with aging [91]. Movement initiation and execution networks undergo a decrease of connectivity between sensorimotor cortex (S1/M1) and striatum while cerebellar connectivity is unchanged [92]. However, the cerebellum participates in motor networks during simple tasks (e.g. elbow flexion-extension) in old subjects, while young subjects perform the task without cerebellar involvement [93]. In rats, global structural and functional network features remain largely similar during aging [94]. However, connectivity decreases in sensory areas, while limbic connectivity increases [95]. During cognitive tasks, old rats engage larger networks compared to young rats [96]. Although locomotion networks have not yet been studied in rats, we can assume from these findings that age-related factors may have contributed to the increase of the vermis network in the chronic stage after 6-OHDA injection."

Reviewer 2 Report

Comments and Suggestions for Authors

In this study, the authors used metabolic connectivity analysis based on [18F]FDG uptake in a unilateral 6-OHDA rat model to compare walking-related functional networks in healthy rats with those in the stable chronic stage after unilateral 6-OHDA injection.

The manuscript is well-written, clearly presenting the findings and thoroughly explaining the proposed analysis and methodology. 

Major comments:

-       The detailed metabolic connectivity analysis described by the authors provides novel insights into the compensatory mechanisms triggered by dopaminergic neuron degeneration. This is a significant contribution to the field. However, the study could be further enhanced by including a quantitative gait/locomotor analysis (e.g., CatWalk) at baseline and after depletion. This addition would strengthen the completeness of the study by providing complementary quantitative gait/locomotor data.

-       While the 6-OHDA toxin is a good model for dopamine depletion, it does not fully replicate the pathophysiology seen in Parkinson's disease. Including more context in the introduction about why other animal models are not suitable for the proposed analysis would help readers understand the validity of the animal model used.

-       Quantifying the extent of dopamine lesions using immunohistochemistry can provide a deeper understanding of lesion severity and strengthen the robustness of the observed findings. 

Minor comments:

-       Please include references regarding the semiquantitative method used to measure the accumulation of the radiolabeled dopamine precursor [18F]FDOPA (lines 99-101). 

-       In the Legend of Figure 1, please include the test used to calculate statistical significance. 

-       In the Legends of Figure 2 (line 146) and Figure 4 (line 255), please specify which correction was used for multiple testing. 

Author Response

Dear reviewer,

we are very grateful for your insights and valuable input to our manuscript. We have tried to address all the issues raised, and we believe that in consequence the paper has been significantly improved.

Please find a point-by-point reply to all comments below.

1) However, the study could be further enhanced by including a quantitative gait/locomotor analysis (e.g., CatWalk) at baseline and after depletion. This addition would strengthen the completeness of the study by providing complementary quantitative gait/locomotor data.

You are right - this would have strengthened the data. Unfortunately, we did not do it, and the rats are long gone. However, in a previous study (Kordys et al., 2017, DOI 10.1186/s13550-017-0317-9) we did Catwalk gait analysis in the same 6-OHDA rat model with similar unilateral dopamine depletion severity ([18F]FDOPA-PET). We found that mainly the contralesional forepaw was affected, with significantly reduced swing speed and significantly increased swing duration. A significantly increased print area of the ipsilesional hindpaw indicated a weight shift to this paw during diagonal gait. We did extensive correlation analyses between dopamine depletion severity and gait parameters, and also between [18F]FDG uptake in different brain areas and gait parameters.

We added the following sentence to paragraph 4.5: "For detailed gait analysis and correlation of gait parameters with dopamine depletion severity (measured with [18F]FDOPA-PET) and metabolic activity (measured with [18F]FDG-PET) in the same rat model see [14]."

2) While the 6-OHDA toxin is a good model for dopamine depletion, it does not fully replicate the pathophysiology seen in Parkinson's disease. Including more context in the introduction about why other animal models are not suitable for the proposed analysis would help readers understand the validity of the animal model used.

We have added the following sentence to the paragraph where we describe the 6-OHDA model in the introduction (lines 65–77) with respect to mild phenotype, behavioral compensation and stable chronic phase: "Due to these characteristics we chose this unilateral model over a bilateral neurotoxin or genetic model, although the genetic models in particular can more accurately replicate Parkinson's disease pathology [15,16]." The two new references are reviews about animal models of Parkinson's disease, Chia et al., 2020 (DOI 10.3390/ijms21072464) and Dovonou et al., 2023 (DOI 10.1186/s40035-023-00368-8).

3) Quantifying the extent of dopamine lesions using immunohistochemistry can provide a deeper understanding of lesion severity and strengthen the robustness of the observed findings.

When we started with the 6-OHDA model and [18F]FDOPA-PET approx. 15 years ago, we indeed did tyrosine hydroxylase immunohistochemistry (TH-IHC) in random samples to confirm that the PET images were reliable, and also included it in publications. Now we don't do confirmational IHC anymore, as the correlation between [18F]FDOPA uptake and TH-IHC is very well established in humans and different animal models including the 6-OHDA rat model. So unfortunately, we did not collect the rats' brains, but let them live until they died of old age. From the PET images we estimated a dopamine depletion in the ipsilesional striatum of approx. 82%.

We have added the following to paragraph 4.6: "As correlation between [18F]FDOPA uptake and dopaminergic cell death is well established [85-88], we did not perform histopathological examinations. For a typical example of tyrosine hydroxylase immunohistochemistry in this model see [19].

4) Please include references regarding the semiquantitative method used to measure the accumulation of the radiolabeled dopamine precursor [18F]FDOPA (lines 99-101). 

We have included the reference Kyono et al., 2011 (DOI 10.1186/2191-219X-1-25), which describes the suitability of the cerebellum as reference region for [18F]FDOPA-PET, as it is not affected in the unilateral 6-OHDA rat model. As a second reference, we included the EANM practice guideline/SNMMI procedure standard for dopaminergic imaging (2020; DOI 10.1007/s00259-020-04817-8). This is focused on human imaging (PET and SPECT) and describes the different methods inluding the semiquantitative one.

5) In the Legend of Figure 1, please include the test used to calculate statistical significance. 

We have added to the figure legend that it was a paired t-test.

6) In the Legends of Figure 2 (line 146) and Figure 4 (line 255), please specify which correction was used for multiple testing. 

We have added to the figure legends that we used the TFCE (threshold-free cluster enhancement) method followed by permutation testing, and provided the reference (Smith and Nichols 2019, DOI 10.1016/j.neuroimage.2008.03.061).